**Data Availability Statement:** All relevant data are within the manuscript.

**Funding:** Aditya Shankar Kataki was supported by Assam Science and Technology Council (ASTEC),

# Evaluation of synergistic effect of entomopathogenic fungi *Beauveria bassiana* and *Lecanicillium lecacii* on the mosquito *Culex quinquefaciatus*

Aditya Shankar Kataki[1,2]*, Francesco Baldini[1,3], Anjana Singha Naorem[2]

**1** School of Biodiversity, One Health, and Veterinary Medicine, University of Glasgow, Glasgow, United Kingdom, **2** Department of Zoology, Cotton University, Guwahati, Assam, India, **3** Environmental Health and Ecological Sciences Department, Ifakara Health Institute, Ifakara, Tanzania

* 2932601K@student.gla.ac.uk

## Abstract

Vector-borne diseases resulted into several cases of human morbidity and mortality over the years and among them is filariasis, caused by the mosquito *Culex quinquefasciatus*. Developing novel strategies for mosquito control without jeopardizing the environmental conditions has always been a topic of discussion and research. Integrated Vector Management (IVM) emphasizes a comprehensive approach and use of a range of strategies for vector control. Recent research evaluated the use of two entomopathogenic fungi; *Beauveria bassiana* and *Lecanicillium lecanii* in IVM, which can serve as potential organic insecticide for mosquito population control. However, their combined efficacy has not yet been evaluated against mosquito control in prior research and a gap of knowledge is still existing. So, this research was an attempt to bridge up the knowledge gap by (1) Assessing the combined efficacy of *Beauveria bassiana* and *Lecanicillium lecanii* on *Culex quinquefasciatus* (2) To investigate the sub-lethal concentration ($LC_{50}$) of the combined fungal concentration and (3) To examine the post-mortem effects caused by the combined fungal concentration under Scanning Electron Microscope (SEM). The larval pathogenicity assay was performed on 4th instar *C. quinquefasciatus* larvae. Individual processed fungal solution of *B. bassiana* and *L. lecanii* were procured and to test the combined efficacy, the two solutions were mixed in equal proportions. To evaluate the sub-lethal concentration ($LC_{50}$), different concentrations of the combined fungal solution were prepared by serial dilations. The mortality was recorded after 24 hours for each concentration. Upon treatment and evaluation, The $LC_{50}$ values of *B. bassiana* and *L. lecanii* were $0.25 \times 10^4$ spores/ml and $0.12 \times 10^4$ spores/ml respectively and the combined fungal concentration was $0.06 \times 10^3$ spores/ml. This clearly indicated that the combined efficacy of the fungi is more significant. Further, SEM analysis revealed morphological deformities and extensive body perforations upon combined fungal treatment. These findings suggested that combining the two fungi can be a more effective way in controlling the population of *Culex quinquefasciatus*.

Guwahati, Assam, India. Francesco Baldini was supported by the Academy Medical Science Springboard Award (ref: SBF007\100094).

# 1. Introduction

## 1.1 Mosquito and their health and economic burden

Vectors are living organisms that are responsible for transferring infectious pathogens between humans or from animals to humans [1] and the diseases that are transmitted by potential vectors (eg: mosquitoes, fleas, cockroaches, tsetse flies, ticks, bugs etc.) are considered as vector-borne diseases [2]. Among vectors, mosquito borne diseases (MBD's) are considered responsible for around 700 million cases of human morbidity and around a million of mortality every year around the globe [2]. *Aedes*, *Anopheles*, and *Culex* are recognized as most important mosquito vectors, responsible for the causing the deadly diseases dengue, malaria, filariasis, West Nile fever, yellow fever, Zika fever and Japanese encephalitis [1, 3]. According to prior research and survey, there are 219 million cases of malaria every year globally and around 3.9 billion are at risk contacting dengue in over 129 countries [1]. Chikungunya and Zika has led to an average yearly loss of over 106,000 and 44,000 disability-adjusted life years (DALYs) worldwide, respectively, between 2010 and 2019 [4, 5]. In addition, there has been significant economic losses too due to mosquito borne diseases (MBD's) worldwide. As per reports, direct global cost of malaria has been estimated to be around \$12 billion per year [6] and the estimated global cost of dengue is US \$306 billion between 2020–2050 [7]. Similarly, mosquitoes of the family Culicidae is distributed worldwide comprising about 3500 species is responsible for high burden nuisance and spread of diseases in any urban cities [8, 9]. The species *Culex quinquefasciatus* is considered as the principal vector of bancroftian filariasis and a potential vector of the disease Dirofilaria immitis [10, 11]. This mosquito species has also been found to be a potential vector of several other arboviruses like avian pox, Rift Valley fever virus and West Nile virus and protozoa like *Plasmodium relictum* that causes bird malaria [12]. However, despite their high nuisance in urban society [8] and being a potential vector of diseases, they remain less studied compared to anophelines or aedes.

## 1.2 Current control methods and drawbacks

Currently various chemical, biological and mechanical methods are implemented to control the mosquito population and mosquito-borne diseases resulting in increase of resistance in them gradually [13–15]. The chemical methods mainly include the use of long-lasting insecticide treated nets (LLINs), indoor residual spraying (IRS) and use of mosquito repellants among others [13] Biological methods mainly include genetic modifications like sterile insect technique (SIT) [13, 16], use of fungi from genera *Beauveria*, *Metarhizium*, *Lagenidium* etc. [17, 18], use of fish, protozoans and bacterial agents. Mechanical control methods mainly involve the use of traps with chemical attractants like $CO_2$, $NH_3$, lactic acid etc that attract female mosquitoes along with eave tubes and attractive sugar baits [19, 20].

However, the chemical insecticides like DDT, pyrethroid, deltamethrin, organochlorine etc are found to be associated with many neurological and immunological disorders in human beings and are also carcinogenic leading to formation of tumors [21]. Additionally, chemical insecticides can affect natural predators, parasitoids and other organisms that assist in natural pest control resulting in harm to a broad variety of insects outside their intended target [22]. The ecosystem, including plant health, soil quality and other organisms that depend on these insects for a variety of ecological services, may be negatively impacted by loss of diversity due to the use of insecticides [22]. Moreover, target pests may become resistant to a particular insecticide after an extended period of exposure making it less effective [15, 23]. Thus, finding alternatives to these chemical insecticides has always been a topic of discussion and research among scientists.

## 1.3 Entomopathogenic fungi- A biological control agent for mosquito control

In search of alternatives, microbial biopesticides have being found to have undergone a great momentum around the globe [22]. Several studies and experiments have demonstrated the potential use of entomopathogenic fungi for controlling mosquito vectors without jeopardizing the nature [24, 25]. It has been well cited in previous research experiments that entomopathogens can contribute to reducing selective pressure for pesticide resistance development in pest populations [26, 27]. *Beauveria bassiana* and *Lecaniciilium lecanii* has been found to play a key role in management and elimination of forestry, veterinary and agricultural pests [22, 28] Particularly in honeybee research, experimental investigations employing entomopathogenic fungi have yielded great success [29]. Thus, the potential use of such entomopathogens to control mosquito population and restrict the spread of mosquito-borne disease has largely driven the research and investigation on fungal-mosquito interactions over more than a century [17, 30–32]. Mosquito larvae are exposed to fungi when they are exposed to plan detritus, within the water column and at the surface of water [30]. Adult mosquitoes are exposed to fungi in both indoor and outdoor environments when they rest, blood feed, mate and oviposit [30]. It has been well cited in prior research and experiments that the fungal infection reduces the life span of insecticide resistant mosquitoes [32–34] and thus entomopathogenic fungi can be used as a synergy in various insecticides or alone in IVM strategies [32].

## 1.4 Mode of action of entomopathogenic fungi

The main feature that makes these entomopathogenic fungi unique is their infectivity through direct contact and penetration [29] *B. bassiana* and *L. lecanii* successfully penetrate the cuticle of the insect and then reach the haemocoel to overcome the hosts innate immune defense response [30]. The conidia or spores germinate on the cuticle of the insect host, penetrate through the cuticle and spread in the hemolymph finally resulting the death of the host [35]. Upon penetration, it employs a combination of biochemical and mechanical mechanisms to infiltrate the host integument and reach the haemocoel [21]. When mycelium reaches a nutrient-rich environment, it switches to a specialized yeast-like cell phenotype known as hyphal bodies or blastophores [36]. and start to exploit the nutrient rich environment of the insect blood which ultimately results in colonizing tissues and release of toxic metabolites. It eventually degrades the free amino acids in the hemolymph and inhibits several key metabolic enzymes, including glutathione S- transferases (GST), carboxylesterase (CarE), and cytochrome P450 (CYP450) [18, 29, 32, 35–37] that ultimately results in death of the host upon fungal infection by mummification and mycosis [29].

Despite studies and research on their effectiveness to use as an organic insecticide and synergically with other insecticides for mosquito population control [38, 39], the combined efficacy of the two entomopathogenic fungi *B. bassiana* and *L. lecacii* has not yet been explored in previous research and experiments and thud needs to be studied and understand. This can indeed serve as an effective IVM strategy to prevent the emergence of insecticide resistance in mosquitoes. Furthermore, till date, no research has been done on assessing the morphological deformities and abnormalities that might have resulted upon combined fungal treatment of *B. bassiana* and *L. lecacii* on mosquitoes. Thus, this research was an attempt to bridge up existing gap of knowledge. The hypotheses of the research were (1) Combining the two entomopathogenic fungi *B. bassiana* and *L. lecanii* will produce better efficacy results than their individual treatments (2) The sub-lethal concentration ($LC_{50}$) of the combined fungal solution will be less than their individual solutions.

To test the hypotheses, the research addresses mainly three objectives: (i) evaluating the synergistic effect of *B. bassiana* and *L. lecanii* and comparing them with individual effect (ii) assessing the sub-lethal concentration ($LC_{50}$) of the combined fungal formulation and individual treatments against *C. quinquefasciatus* (iii) examining the post-mortem effects and morphological deformities on fungal infected larvae under SEM and compare it with the untreated larvae.

## 2. Materials and methods

### 2.1 Institutional Ethical Clearance (IEC)

The present work involved the use of mosquitoes but not any human. So, Institutional Ethical Clearance certificate of approval was obtained from Institutional Biosafety Committee (IBSC) of Cotton University, Guwahati, Assam; India with reference no–CU/IBSC/2023/21 dated on 9th March 2023.

### 2.2 Collection sites of *Culex quinquefasciatus* larvae samples

The 4th instar larval samples of *C. quinquefasciatus* were collected during the pre-monsoon period (March- April'23) from different locations in Guwahati Metropolitan city, Kamrup (M) district, Assam (26.1158˚ N, 91.7086˚ E) as shown in [Fig 1]. The selection of the sample collection sites was based on the factor that despite high population density of *C. quiquefasciatus* [9] and residents in these urban areas, no relevant survey or experimental study on mosquitoes till date were conducted.

### 2.3 Experiment location

The entire experiment was carried out in the laboratory of Zoology department, Cotton University, Guwahati, Assam, India. The laboratory condition was at temperature 25˚C to 37˚C and humidity 70% to 80% with a 12-hour day/night cycles.

### 2.4 Larvae collection and laboratory rearing

The mosquito larvae from different sites were collected using a larval collection dipper and scoopers as per standard protocols [40, 41]. Upon collection, the mosquito larvae were transferred in small containers covered with lids for identification before rearing. The larvae were identified using the books entitled "The Ecology of Malaria mosquitoes" by Charlwood JD and "The biology of mosquitoes" by A.N Clements [42, 43]. Upon identification, only larvae of *C. quinquefasciatus* were selected and others were discarded. The larvae were transferred to trays of 5" x 7" in dimension for larval bioassay test. The mosquito larvae were fed with dog biscuits and millet powder and yeast powder in 3:3:1 ratio as done in prior research [21] during the experiment.

### 2.5 Preparation of different concentrations of *B. bassiana* fungal solution

Processed fungal solution of *B. bassiana* with concentration $2 \times 10^8$ spores/ml was procured from the S.S. Biotech brand of Guwahati, Assam, India. This was considered as stock concentration for *B. bassiana* fungal formulation. The viability of the spores was determined by culture on nutrient agar and counting the colonies formed. Further, serial dilations were performed upon the stock concentration in 1:10 dilution factor by following serial dilution protocol [44] and different concentrations of *B. bassiana* fungal formulation were prepared to evaluate the sub-lethal concentration ($LC_{50}$) against *C. quinquefasciatus* larvae. Upon preparation they were mixed in the larval pans. The different fungal concentrations of *B. bassiana*

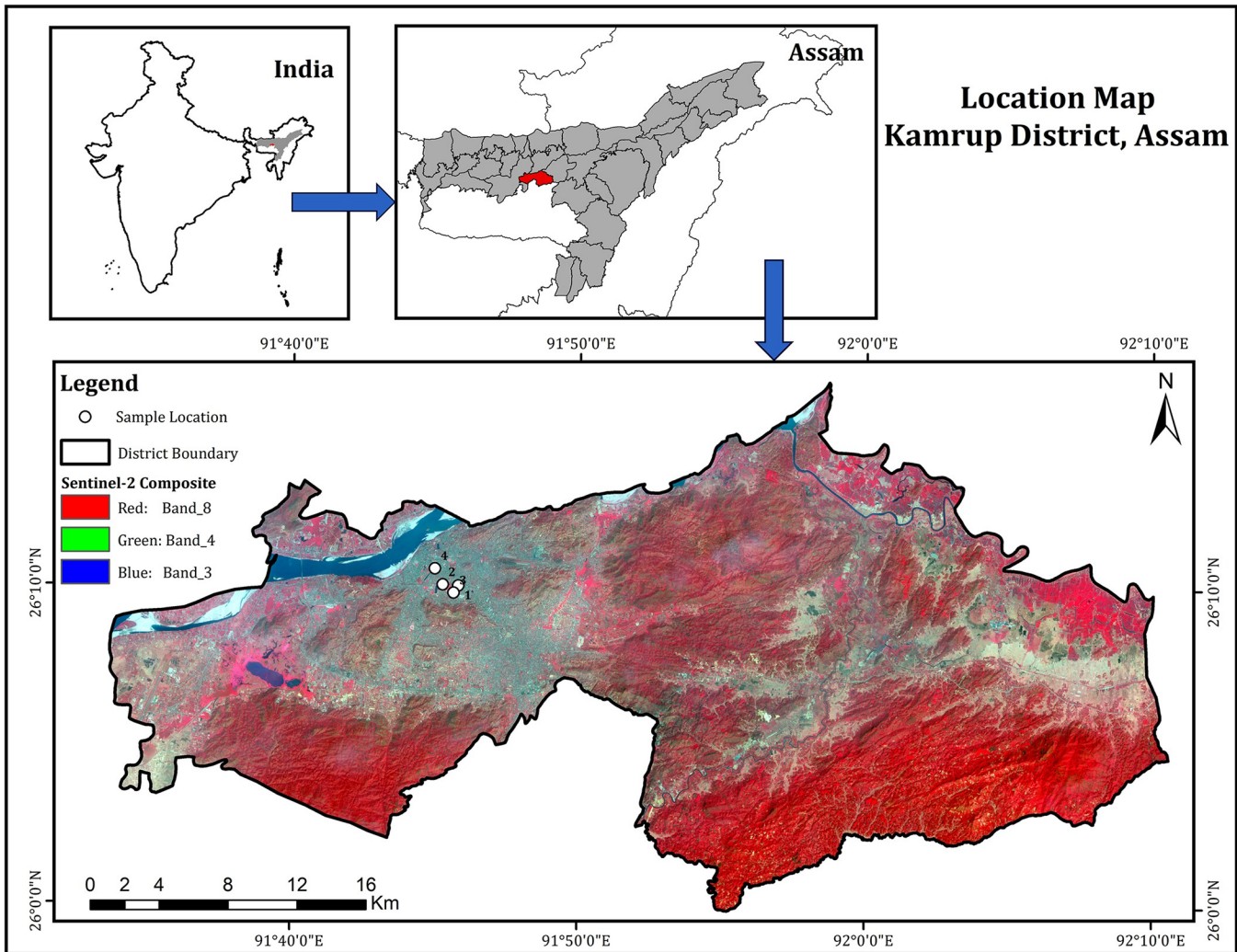

**Fig 1. Sentinel-2 data was downloaded from USGS Earth Explorer portal (https://earthexplorer.usgs.gov/).** Three bands-Band 8 (Near Infrared), Band 4 (Red) and Band 3 (Green) were used to prepare the False Colour Composite (FCC) Image. This was performed using Composite Band Tool in Arc GIS 10.2 software. In FCC image, vegetation appears in red, water bodies are shown in dark blue, and built-up areas are depicted in light blue. Barren land, rock, outcrops, and sandbars are represented in shades ranging from light blue to grey.

prepared were 2.5 x 10$^6$ spores/ml, 1.5 x 10$^4$ spores/ml, 1.25 x 10$^4$ spores/ml, 0.52 x 10$^4$ spores/ml, 0.25 x 10$^3$ spores/ml and 0.12 x 10$^3$ spores/ml. The counting of the spores was carried out using hemocytometer.

## 2.6 Preparation of different concentrations of *L. lecanii* fungal solution

Processed fungal solution of *L. lecanii* with concentration 1 x 10$^9$ spores/ml was procured from the S.S. Biotech brand of Guwahati, Assam, India. This was considered as stock concentration for *L. lecanii* fungal formulation and similarly the viability of the spores was determined by culture on nutrient agar and counting the colonies formed. Further, serial dilations were performed serial dilations were performed upon the stock concentration in 1:10 dilution factor by following serial dilution protocol [44] and different concentrations of *L. lecanii* were prepared to evaluate the sub-lethal concentration (LC$_{50}$) against *C. quinquefasciatus* larvae. Similarly, upon preparation, they were mixed in the larval pans. The different fungal

concentrations of *L. lecanii* prepared were $1 \times 10^5$ spores/ml, $0.5 \times 10^5$ spores/ml, $0.25 \times 10^4$ spores/ml, $0.12 \times 10^4$ spores/ml, $0.06 \times 10^4$ spores/ml and $0.03 \times 10^4$ spores/ml. The counting of the spores was carried out using hemocytometer.

## 2.7 Preparation of combined fungal solution of *B. bassiana* and *L. lecanii*

A stock solution of 100 ml, combining two entomopathogenic fungi *B. bassiana* and *L. lecanii* was prepared by adding 50 ml each of the individual fungal solution. The concentration of the stock combined fungal solution was $6 \times 10^{10}$ spores/ml and was kept in a 250 ml conical flask covered with aluminum foil to avoid contamination at 27˚C room temperature. To evaluate the sub-lethal concentration ($LC_{50}$) the combined fungal solution, serial dilations were performed similarly upon the stock concentration in 1:10 dilution factor by following serial dilution protocol [44]. After performing serial dilations, the combined fungal concentration procured were $0.5 \times 10^4$ spores/ml, $0.25 \times 10^4$ spores/ml, $0.12 \times 10^4$ spores/ml, $0.06 \times 10^3$ spores/ml, $0.03 \times 10^3$ spores/ml and $0.02 \times 10^3$ spores/ml.

## 2.8 Larval pathogenicity bioassay

The larval pathogenicity bioassay was carried out using standard mosquito larval bioassay protocols [45] for the individual fungal solution of *B. bassiana*, *L. lecanii* and for the prepared combined fungal solution. The experiment was conducted by exposing larvae in conical flask of 250 ml. Three conical flasks for each; Control (distilled water), *B. bassiana* fungal solution, *L. lecanii* fungal solution and combined fungal solution was taken. The three conical flasks were considered as three replicates for each solution. Around twenty larvae were transferred and exposed in each replicate. Upon exposure and observation, larval mortality was noted down after 24 hours as mentioned by Standard protocols for larval bioassay and done in other studies too [45–47]. The mortality was corrected using Abbott's formulae of corrected mortality [48, 49].

## 2.9 Preparation of larvae sample for SEM analysis

After treatment with the fungal solution the larvae specimens were fixed with 2.5% glutaraldehyde for 24 hours followed by dehydration in a series of acetone solutions (30%, 50%, 70%,90%) for 10 mins in each set and finally in 100% for 5 mins. Then the larval sample was air dried, and sputter coated with a 45nm gold coated and was observed under SEM (Sigma VP) [50]. Observations were mainly made on the head and thoracic regions of control and treated larvae, the abdominal segments and in the terminal region of larvae showing respiratory siphon for comparison studies.

## 2.10 Statistical analysis

Statistical analysis was performed in RStudio software (version 2024.04.2+764) and the packages 'ggeffects' 'ggplot2' 'lme4' 'Matrix' and 'lmerTest' using generalized linear mixed-effects model (GLMM) to investigate the significance of several explanatory variables 'fungi', 'concentration' and their interaction 'fungi * concentration' on the response variable 'mortality' caused by individual and combined fungal solution. The model used binary distribution and the final simplified model was selected based on AIC. The final model was plotted using the "ggplot2" package of the RStudio software. Additionally, Abbotts formulae was calculated by = (% test mortality -% control mortality/ 100—control mortality x 100) [49] and probit analysis was done using tables to estimate the probits and fitting the relationship by eye.

**Table 1. Larval mortality observed post 24 hours of treatment with *B. bassiana* fungal solution.**

| Sl. No. | Concentration of fungal solution (spores/ml) | Log concentration (spores/ml) | Mean Mortality ±S. D | C.M % (Abbott's formulae) | Probit (P) |
|---|---|---|---|---|---|
| | Control (Distilled water) | | 8.34±2.89 | 9 | 3.66 |
| 1 | $2.50 \times 10^6$ | 6.39 | 92.34±2.52 | 91.64 | 2.63 |
| 2 | $1.50 \times 10^4$ | 6.17 | 85.67±2.08 | 84.36 | 2.56 |
| 3 | $1.25 \times 10^4$ | 4.09 | 74±4 | 71.63 | 1.51 |
| 4 | $0.52 \times 10^4$ | 3.71 | 53.67±1.53 | 49.45 | 1.38 |
| 5 | $0.25 \times 10^3$ | 3.41 | 47.67±2.51 | 42.9 | 1.26 |
| 6 | $0.12 \times 10^3$ | 2.07 | 40±2 | 34.54 | 0.77 |

## 3. Results

### 3.1 Larval mortality upon treatment with *B. bassiana* fungal solution

Upon treatment with different concentrations of *B. bassiana* fungal solution, it was observed that the sub-lethal concentration ($LC_{50}$) for it was around $0.52 \times 10^4$ spores/ml (Table 1). Larvae that were exposed in concentration $2.5 \times 10^6$ spores/ml was giving mean morality of 90% and thus it was considered as $LC_{90}$ concentration.

### 3.2 Larval mortality upon treatment with *L. lecanii* fungal solution

Upon treatment with different concentrations of *L. lecanii* fungal solution, it was observed that the sub-lethal concentration ($LC_{50}$) for it was around $0.12 \times 10^4$ spores/ml (Table 2). Larvae that were exposed in concentration $1 \times 10^5$ spores/ml was giving mean morality of around 87–90% and thus it was considered as $LC_{90}$ concentration.

### 3.3 Larval mortality upon treatment with different concentrations of combined fungal solution of *B. bassiana* and *L. lecanii*

After treatment with different concentrations combined fungal formulation of *B. bassiana* and *L. lecanii*, it was observed that the sub-lethal concentration ($LC_{50}$) for it was around $0.06 \times 10^3$ spores/ml (Table 3). Larvae that were exposed in concentration above $1 \times 10^5$ spores/ml was giving mean morality of around 90–95% and thus it was considered as $LC_{90}$ concentration.

Upon performing statistical analysis, it was found that the response variable "mortality" was significantly associated an interaction between fungal infection (individual species and combined) and their concentration with **chisq ($\chi^2$) = 16.579, Df = 2 and Pr(>Chisq) = 0.0002511 \*\*\*.**

Thus, the sublethal ($LC_{50}$) concentration of individual fungal solutions of *B. bassiana* and *L. lecanii* were $0.52 \times 10^4$ spores/ml and $0.12 \times 10^4$ spores/ml respectively. However, when

**Table 2. Larval mortality observed post 24 hours of treatment with *L. lecanii* fungal solution.**

| Sl. No. | Concentration of fungal solution (spores/ml) | Log concentration (spores/ml) | Mean Mortality ±S. D | C.M % (Abbott's formulae) | Probit (P) |
|---|---|---|---|---|---|
| | Control (Distilled water) | | 8.34±2.89 | 9 | 3.66 |
| 1 | $1 \times 10^5$ | 5.00 | | 87.27 | 2.51 |
| 2 | $0.5 \times 10^5$ | 4.69 | 88.34± 1.52 | 82.54 | 2.31 |
| 3 | $0.25 \times 10^4$ | 3.39 | 84±1 | 58.91 | 1.52 |
| 4 | $0.12 \times 10^4$ | 3.07 | 62.34±2.51 | 50.18 | 1.27 |
| 5 | $0.06 \times 10^4$ | 2.78 | 54.34±2.08 | 42.24 | 1.08 |
| 6 | $0.03 \times 10^4$ | 2.47 | 42.34±2.52 | 26.18 | 0.84 |

**Table 3. Larval mortality observed post 24 hours of treatment with combined fungal solution of *B. bassiana* and *L. lecanii*.**

| Sl. No. | Concentration of fungal solution (spores/ml) | Log concentration (spores/ml) | Mean Mortality ±S. D | C.M % (Abbott's formulae) | Probit (P) |
|---|---|---|---|---|---|
| | Control (Distilled water) | | 8.34±2.88 | 9 | 3.66 |
| 1 | $0.5 \times 10^4$ | 3.69 | | 78.18 | 1.31 |
| 2 | $0.25 \times 10^4$ | 3.39 | 80±5 | 76.36 | 1.12 |
| 3 | $0.125 \times 10^4$ | 3.09 | 78.34±7.63 | 76.72 | 0.91 |
| 4 | $0.06 \times 10^3$ | 2.77 | 78.67±6.50 | 47.99 | 0.66 |
| 5 | $0.03 \times 10^3$ | 2.47 | 52.33±2.51 | 18.17 | 0.38 |
| 6 | $0.015 \times 10^3$ | 2.17 | 25±5 | 3.63 | 0.10 |

combined, the $LC_{50}$ value was $0.06 \times 10^3$ spores/ml indicating better efficacy and synergic effect against *C. quinquefasciatus* mosquito species as shown by graphical representation in [Fig 2].

### 3.4 Depositional of fungal spores in the larvae upon combined fungal treatment at different concentrations

Upon application of the combined fungal solution of *B. bassiana* and *L. lecanii* in different concentrations to the *C. quinquefasciatus* larvae, different degrees of deposition of fungal spores were observed under stereo microscope at magnification 40X for each concentration [Fig 3(A)–3(H)]. When carefully observed, the larvae kept in control (distilled water) had no

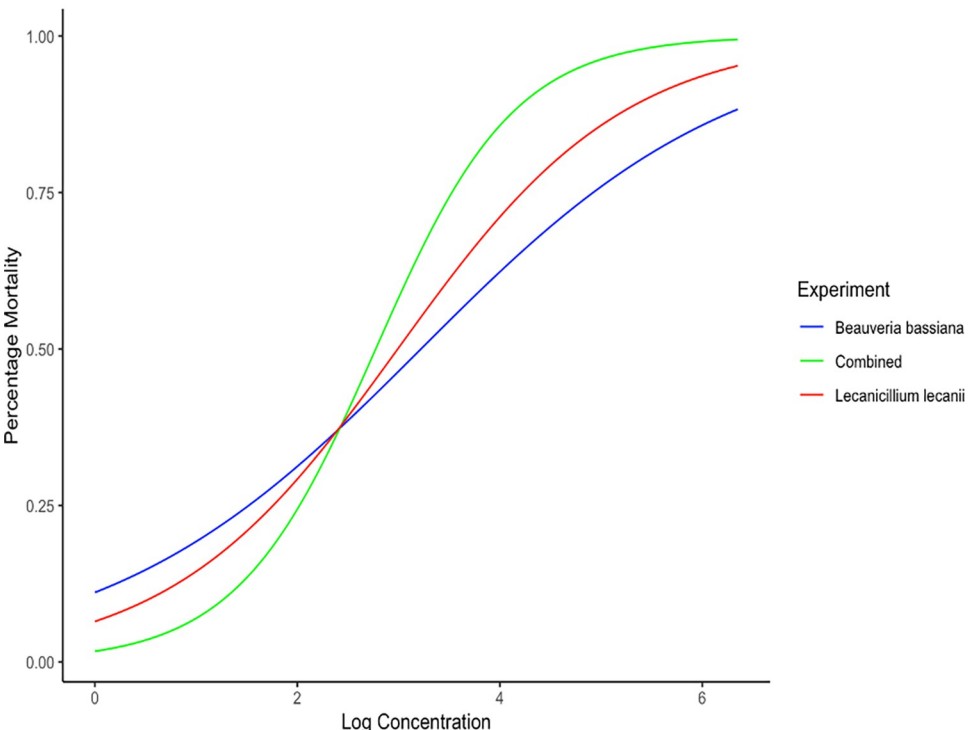

**Fig 2. Percentage mortality of *C. quinquefasciatus* larvae at different concentrations (log transformed) of individual and combined fungal solution.** The X- axis of the graph represents the "Log concentration" and the Y-axis represents the "Percentage Mortality". From the graph the sigmoid curve for combined effect produces 50% ($LC_{50}$) mortality in *C. quinquefasciatus* at much lesser concentration than the two individual fungi.

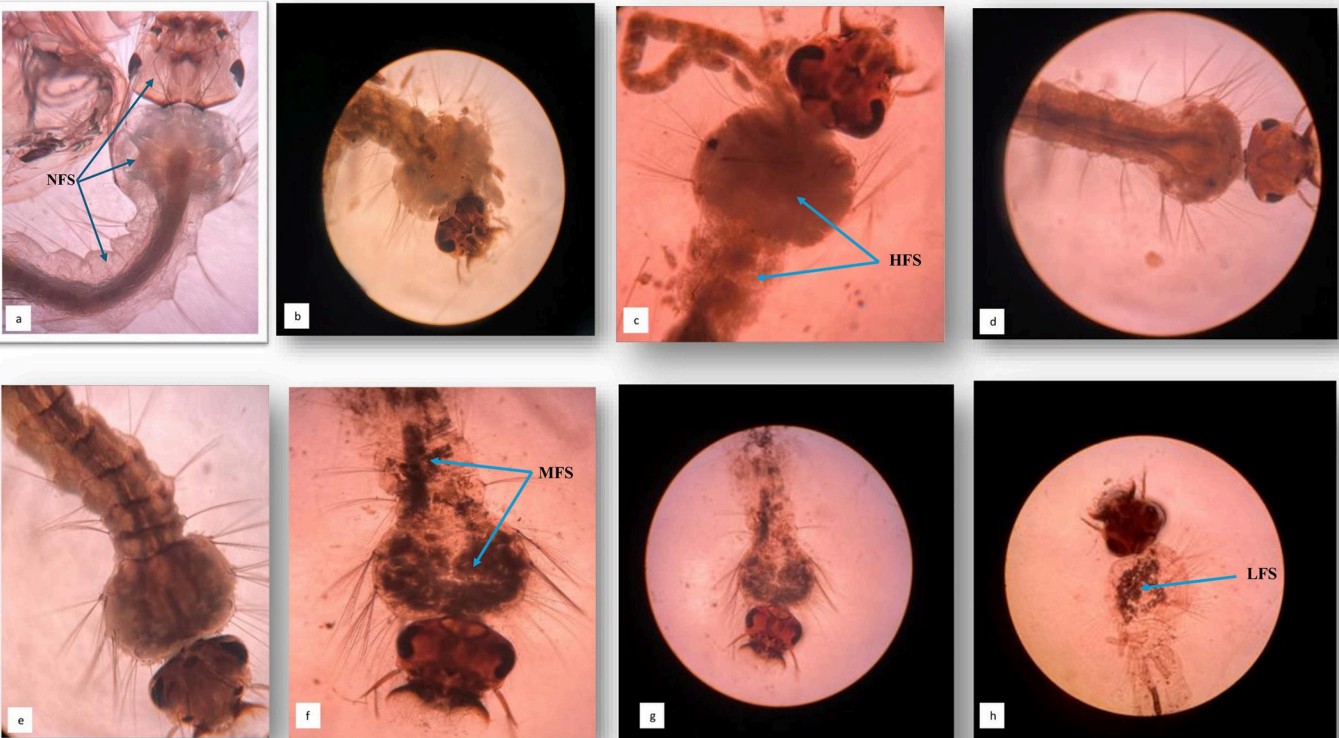

**Fig 3. Images of larvae of *C. quinquefasciatus* under stereo microscope at magnification 40X to show deposition of fungal spores upon treatment with different concentrations of combined fungal solution of *B. bassiana* and *L. lecanii* and compare it with the control untreated larvae.** [a] control (distilled water) with no fungal spores (NFS) [b] $5 \times 10^7$ spores/ml and [c] $2.85 \times 10^6$ spores/ml with high fungal spores (HFS) [d] $1.25 \times 10^5$ spores/ml [e] $1 \times 10^5$ spores/ml and [f] $0.25 \times 10^4$ spores/ml with moderate fungal spores (MFS) and [g] $0.0625 \times 10^3$ spores/ml and [h] $0.31 \times 10^3$ spores/ml with low fungal spores (LFS).

deposition of fungal spores in any region of the body. Additionally, the alimentary canal along with the cuticle also seemed to be intact and complete with no distortion and breakage [Fig 3(A)]. On the contrary, larvae exposed in higher concentrations of combined fungal solution were found to have extensive deposition of fungal spores in them [Fig 3(B) and 3(C)]. These larvae were observed to had completely distorted thorax and alimentary canal with deposition of fungal spores. As the concentrations were gradually decreased, the deposition of fungal spores was also observed to get reduce and the alimentary canal and cuticle were intact [Fig 3(D)–3(F)]. However, the thorax and head regions were found to be swollen up with deposition of fungal spores. Finally, at the lowest concentration, distortion was evidently less, and the alimentary canal and cuticle was found to be completely intact, with minimal deposition of fungal spores in the head and thoracic region of the larvae [Fig 3(G) and 3(H)]. A closer view of the abdominal segment (AB) of treated larvae with combined effect of the entomopathogenic fungi *B. bassiana* and *L. lecannii* showing deposition of fungal spores has been captured under stereo microscope and is shown in [Fig 4].

## 3.5 Observations under Scanning Electron Microscope (SEM)

Upon treatment of larvae of *C. quinquefasciatus* with the sub-lethal concentration ($LC_{50}$) combined fungal solution of *B. bassiana* and *L. lecanii*, mosquito larvae were visualized under the Scanning Electron Microscope (SEM) to observe the extent of morphological deformities caused. The (Fig 5), represents the pictures of control untreated larvae (left) and the treated larvae (right). Upon our observation and careful analysis, it was found that the control larvae

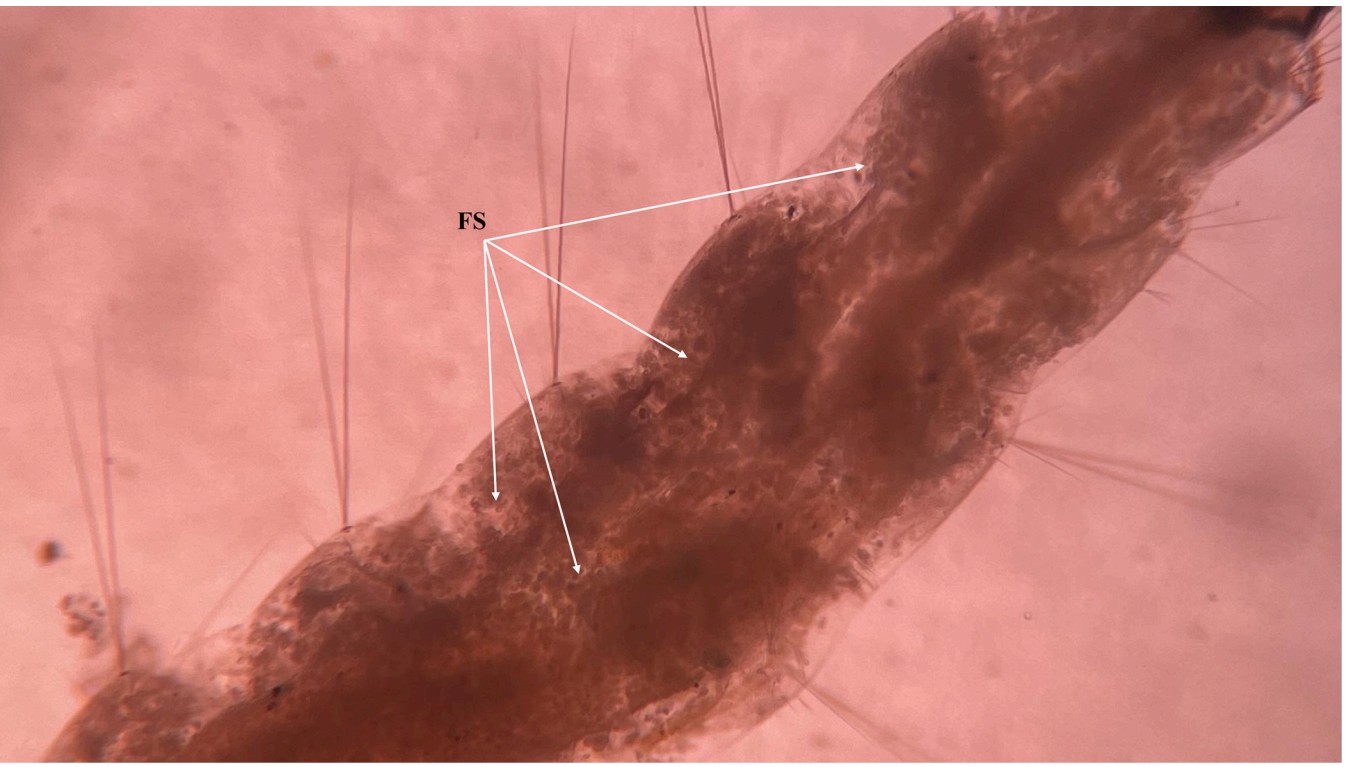

**Fig 4. A closer view of the abdominal segment (AB) of treated larvae with combined effect of the entomopathogenic fungi *B. bassiana* and *L. lecannii* showing deposition of fungal spores.**

have a smooth cuticle surface [Fig 5(A)] whereas treated larvae [Fig 5(B)] had varied level of shrinkage and ruptured cuticle throughout the body.

Additionally, when the cuticle of both the control and treated larvae was analyzed more closely, a complete distortion and aberration of the larval abdominal cuticle was observed [Fig 5(D) and 5(G)] in the treated larvae compared to the control larvae [Fig 5(C)]. The terminal segments were found to be completely shrunken in the treated larvae [Fig 5(F) and 5(H)] which was not actually observed in the control untreated larvae [Fig 5(E)]. Further, abnormalities observed in the treated larvae include breakage in the larval epithelium and bloated thoracic regions.

TFDFDRSTABRSABFDRS

## 4. Discussion

The rising concern of insecticide resistance gradually over the years in mosquitoes around the globe has led researchers to think about other sustainable alternatives for mosquito population control [15, 51–53]. As a result, the use of entomopathogenic fungi alone and synergically with other insecticides has been mentioned in many prior experiments and research work as alternatives to artificial chemical insecticides for mosquito population control and other agricultural pests too [17, 25, 27, 29, 32, 35, 37, 38, 54, 55]. Also, the matter of fact that these entomopathogenic fungi are safe with minimal risks to humans, animals and the environment, makes them a better alternative to the existing chemical artificial insecticides like DDT, Permethrin, Deltamethrin, Organochlorines, Organophosphates etc for mosquito population control [56–58].

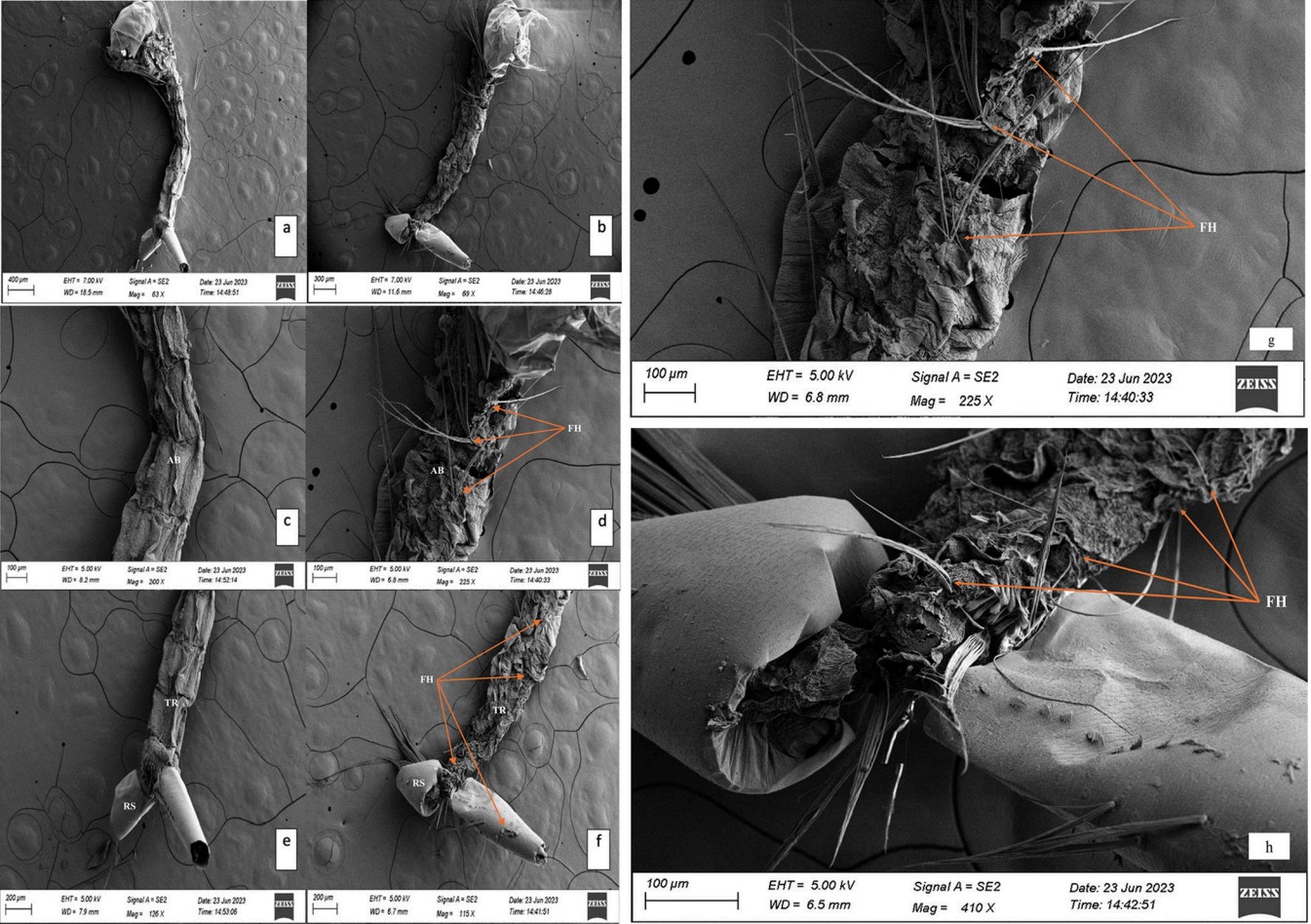

**Fig 5.** Scanning Electron Microscope (SEM) images of *C. quinquefasciatus* larvae of control (left) and combined fungal treated larvae (right) showing deformity and fungal hyphae (FH). Full view of control (a) and treated larvae (b). Abdominal segments (AB) of control (c) and treated (d and g) larvae. The terminal region (TR) of larvae showing respiratory siphon (RS) of control (e) and treated (f and h) larvae.

However, previous studies didn't evaluate the combined efficacy of the two entomopathogenic fungi *B. bassiana* and *L. lecanii* against mosquito population control. In our research we combined the two fungal solutions of *B. bassiana* and *L. lecanii* and found that by combining the fungal solution, it becomes more virulent than the individual fungi against the mosquito species *C. quinquefasciatus*. The viability of spores was determined by culture on nutrient agar and counting the colonies formed. The $LC_{50}$ values for *B. bassiana* and *L. lecanii* were 0.52 x $10^4$ spores/ml and 0.12 x $10^4$ spores/ml respectively and when they were combined in equal proportions, the $LC_{50}$ value was 0.06 x $10^3$ spores/ml against *C. quinquefasciatus* upon exposure for 24 hours. Thus, according to our results and observation, the sub-lethal concentration ($LC_{50}$) of the combined entomopathogenic fungi was ten times more virulent than the individual fungal solutions indicating better efficacy in population control of mosquito species *C. quinquefasciatus*.

In addition, the SEM analysis of the treated larvae with combined fungal solution revealed shrinkage and deformities in the head, thorax and abdomen of the mosquito. Hence, it established the fact that, when the two entomopathogenic fungi were combined at such low concentration, they were able to cause fungal infection and deformities in the larvae leading them to death by mycosis and mummification.

The sub-lethal concentration ($LC_{50}$) value is defined as the concentration of a chemical that kills 50% of the test animals exposed to the concentration for a set period [59]. This is the concentration that is taken into consideration while making any commercial insecticide alone or mixing with other insecticides because animal or insect toxicity studies do not necessarily extrapolate to humans [60, 61]. Thus, it is important to know what the sub-lethal (LC50) concentration or dose of any new insecticide or fungicide is while developing them before using them commercially against insects and pests in large scale so that it doesn't lead to human infection and diseases.

The sub-lethal concentration ($LC_{50}$) evaluated from the current research for the combined fungal solution of *B. bassiana* and *L. lecanii* against *C. quinquefasciatus* mosquito species, can be used as a reference for producing other organic sustainable novel insecticides. The combined fungal solution can be either applied alone or synergically with other insecticides as done with different group of fungi in prior research [38, 39] for controlling different vectors responsible for causing diseases in human and wildlife. Keeping in mind the morphological deformities shown in the experiment through SEM analysis, the concentration can either be increased or decreased for other insects or species of mosquito in different parts of the world where the resistance status of them varies accordingly. This will ensure that their population is below the threshold level and consequently people contaminate with less vector-borne diseases. Apart from that, this combined fungal formulation of *B. bassiana* and *L. lecanii* can be used in local sewages and stagnant water bodies too which are considered as potential breeding sites and habitats for mosquitoes. These will ensure the mosquitoes contaminate themselves with the fungal infection in their larval stage itself and not reach the pupal stage of development, ultimately decreasing the adult population of the mosquitoes in the community.

However, more research is required to understand and determine whether it can be used as aerosols to eradicate other harmful pests and insects.

Since, in recent studies and research, the use of entomopathogenic fungi has drawn a lot of attention [17, 18, 27, 37, 55], researchers and toxicologists may use this information as a starting point to investigate the effects of the fungal solution on different organisms or to compare the toxicity of different substances. This will help them in understanding and establishing safe exposure limits along with developing appropriate sustainable mitigation strategies for vector control in society.

## 5. Conclusion

The present study was an attempt to show the combined efficacy of the two entomopathogenic fungi *B. bassiana* and *L. lecanii* against *C. quinquefasciatus* population control. The research addressed significantly the fact that when the fungi are combined, they indeed show better efficacy than the two individual fungi in controlling population of *C. quiquefasciatus*. In addition, it can be used as a reference in developing alternative organic insecticides for tackling the developing insecticide resistance in mosquitoes around the globe to control their population and spread of vector-borne disease. Thus, the study will be instrumental for producing other insecticides and controlling potential vectors. Moreover, by determining the lethal concentration further for aquatic organisms or invertebrates, it will help evaluate the risk posed by the fungal solution to them. This information thus, will indeed aid in future decision making and strategies in IVM and sustainable vector control in society.

## Acknowledgments

The authors would like to thank the Institute of Advanced Study in Science and Technology (IASST), Guwahati, Assam, India for their support in performing SEM analysis. Moreover, we

would also like to thank the Zoology department of Cotton University, Guwahati, Assam, India and School of Biodiversity, One Health and Veterinary Medicine (SBHOVM) of the University of Glasgow, United Kingdom. Also, the first author would like to thank Dr Nameirakpam Nirjanta Devi; Biotechnology department, Cotton University, Guwahati, Assam, India, for her valuable suggestions and insights.

## Author Contributions

**Conceptualization:** Aditya Shankar Kataki.

**Data curation:** Aditya Shankar Kataki.

**Formal analysis:** Francesco Baldini.

**Investigation:** Anjana Singha Naorem.

**Methodology:** Aditya Shankar Kataki.

**Software:** Aditya Shankar Kataki.

**Supervision:** Anjana Singha Naorem.

**Visualization:** Anjana Singha Naorem.

**Writing – original draft:** Aditya Shankar Kataki.

**Writing – review & editing:** Francesco Baldini.

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
