## [Decision Letter · Decision Letter 0]

28 Jun 2024

PONE-D-24-10887EVALUATION OF SYNERGISTIC EFFECT OF ENTOMOPATHOGENIC FUNGI BEAUVERIA BASSIANA AND LECANICILLIUM LECANII ON THE MOSQUITO CULEX QUINQUEFASCIATUSPLOS ONE

Dear Dr. Kataki,

Thank you for submitting your manuscript to PLOS ONE. After careful consideration, we feel that it has merit but does not fully meet PLOS ONE’s publication criteria as it currently stands. Therefore, we invite you to submit a revised version of the manuscript that addresses the points raised during the review process.

Dear AuthorThere is 2 differents reviewers decisions one as a major and the 2nd reject the work. Please give a special attention to your next revision, and submit a carefull responses to reviewers (especially reviewer 2).Good luck

We look forward to receiving your revised manuscript.

Kind regards,

Rachid Bouharroud

Academic Editor

PLOS ONE

 [FB was supported by the Academy Medical Science Springboard Award (ref: SBF007\\100094)].  

4. We note that Figures 1,2, and 4 in your submission contain [map/satellite] images which may be copyrighted. All PLOS content is published under the Creative Commons Attribution License (CC BY 4.0), which means that the manuscript, images, and Supporting Information files will be freely available online, and any third party is permitted to access, download, copy, distribute, and use these materials in any way, even commercially, with proper attribution. For these reasons, we cannot publish previously copyrighted maps or satellite images created using proprietary data, such as Google software (Google Maps, Street View, and Earth). For more information, see our copyright guidelines: http://journals.plos.org/plosone/s/licenses-and-copyright.

1. You may seek permission from the original copyright holder of Figures 1,2, and 4  to publish the content specifically under the CC BY 4.0 license.  

Additional Editor Comments (if provided):

Reviewers' comments:

Reviewer's Responses to Questions

**Comments to the Author**

1. Is the manuscript technically sound, and do the data support the conclusions?

Reviewer #1: Yes

Reviewer #2: No

2. Has the statistical analysis been performed appropriately and rigorously? 

Reviewer #1: Yes

Reviewer #2: No

3. Have the authors made all data underlying the findings in their manuscript fully available?

Reviewer #1: Yes

Reviewer #2: Yes

4. Is the manuscript presented in an intelligible fashion and written in standard English?

Reviewer #1: Yes

Reviewer #2: No

5. Review Comments to the Author

Reviewer #1: This study explored the effect of applying a mixture of two entomopathogenic fungi to Culex quinquefasciatus larvae. Overall, I think the study was well-performed, analyzed, and presented. The writing would benefit from additional proofreading, as there were a number of typos and sentences in need of corrections. Additionally, it would be beneficial to closely look at the references cited for various claims, as not all of them appeared to be appropropriate at first glance. For instance the mosquito keys referred to for larval IDs specifies a paper looking at adult morphology. On Line 51 statements regarding health effects of pesticides are supported only with another paper reporting on fungi - given that these topics can be quite controversial, one should really cite the original supporting literature.

My other concern relates to the details of the applications. How exactly were the spores formulated, just dry or in oil, or something else? And how were they applied to the larval pans, to the surface of the water or mixed in? Finally, I’m a bit confused regarding the timing - 24 hours seems like a very short time to see such strong effects on survival, I would have expected this to take several days. What was the reason for keeping the assay for one day only? This is something that should be explored in the discussion and the results here compared to that in other similar studies.

Reviewer #2: Comments to the authors

The manuscript titled “EVALUATION OF SYNERGISTIC EFFECT OF ENTOMOPATHOGENIC FUNGI BEAUVERIA BASSIANA AND LECANICILLIUM LECANII ON THE MOSQUITO CULEX QUINQUEFASCIATUS”. The present study describing to evaluate 1) the synergistic effect of the two entomopathogenic fungi Beauveria bassiana and Lecanicillium lecanii and compare them with individual effect; 2) to evaluate the lethal concentration (LC50) of the combined effect of the two fungi regarding the mortality of Culex quinquefasciatus and 3) to examine the post- mortem effects and morphological deformities on fungal infected larvae under Scanning 108 Electron Microscope (SEM) and compare them with the control untreated larvae. After carefully reviewing this manuscript, I can not see some merit in this research. The current form contains several fundamental, technical, grammatical and typographical errors so please carefully revise the entire manuscript.

Major errors

1.In the abstract the objective and methodology are not clear and hard to understand so revise it carefully.

2.Line 22: cite reference.

3.The introduction is poorly written, the current form does not meet scientific standards. Also, only 14 references have been provided, most of the sentence the citation is missing.

Write introduction with following information;

a. Mosquitoes and their health and economic burden

b. Current control methods and their drawbacks

c. Biological control agent and entomopathogenic fungi mediated mosquito control and your objectives

4.The current form of the introduction of several scientific information is missing so carefully revise it.

5.Line 122: typo error

6.Line 131: dog biscuit? dog biscuit WHO recommended one? Why have you used this diet? Is there any special reason?

7.Line 144-145: how did you prepare for test concentration?

8.Line 175: Toxicity or pathogenicity? Confirm it.

9.The methodological part is poorly written and contains several pieces of information.

10.The results are poorly written and data interpretation hard to follow so carefully re-edit it.

11.For discussion two paragraphs sufficient for a scientific paper? Did you use any new reference added in the discussion part apart from introduction and methodology? The discussion part is poorly discussed.

12.The conclusion needs more clear information; the current form is not sufficient.

13.The reference part contains several formation and typographical errors so carefully revise it.

14.Figure 1 does not provide clear and high quality of images.

15.Figure 2: I can not see any damages and any information from this figure so provide clear images.

16.In the SEM analysis I can not see any entomopathogenic fungi spores and conidia but does the author see any spores?

17.Throughout the manuscript I can see several grammatical and typographical errors so carefully fix it.

6. PLOS authors have the option to publish the peer review history of their article (what does this mean?). If published, this will include your full peer review and any attached files.

Reviewer #1: No

Reviewer #2: No

---

## [Author Response · Author response to Decision Letter 0]

22 Jul 2024

Date: 22.07.2024

To The Academic Editor,

PLOS One Journal.

This is to kindly intimate that I, Aditya Shankar Kataki, the first and corresponding author of the submitted manuscript titled “Evaluation of synergistic effect of entomopathogenic fungi Beauveria bassiana and Lecanicillium lecacii on the mosquito Culex quinquefaciatus” with the help of my esteemed co-authors, Dr Francesco Baldini and Dr Anjana Singha Naorem, has carefully revised our work and manuscript once more, considering the feedback provided by esteemed reviewers.

Keeping in mind the reviews of reviewer 1 and reviewer 2, we have incorporated the following changes, and the manuscript was once again thoroughly revised. Also, rigorous statistical analysis was performed using RStudio software (version 2024.04.2+764) and the packages ‘ggeffects’ ‘ggplot2’ ‘lme4’ ‘Matrix’ and ‘lmerTest’ using generalized linear mixed-effects model (GLMM). All the additional statistical information has been drafted under the ‘statistical analysis’ subsection of Material and Methodology section. Additionally, the [Fig 1] of the manuscript consisting of (Maps) has now been redesigned from USGS Earth Explorer portal (https://earthexplorer.usgs.gov/) to avoid copyright issues as suggested.

Regarding the funding disclosure, Francesco Baldini (FB) was supported by the Academy Medical Science Springboard Award (ref: SBF007\\100094)] and Aditya Shankar Kataki (ASK) was supported by Assam Science and Technology Council (ASTEC). FB played a crucial role in reviewing of the manuscript, decision to publish and preparation of the manuscript. The fund received by ASK was used in study design, data collection and analysis.

Thus, I would kindly like to intimate respected academic editor, reviewer 1 and reviewer 2 that, the study has been caried out with full generosity and upon extensive literature review for accurate results. Hence, please kindly consider the following revisions and allow our manuscript to get accepted in PLOS One journal for publication.

Thank You.

With regards,

Aditya Shankar Kataki

Masters in research, Ecology and Environmental Biology (SBOHVM)

University of Glasgow, United Kingdom

Response to Reviewers

Reviews of Reviewer 1:

Comment 1.1. How exactly were the spores formulated just dry or in oil?

Response 1.1. The information has been mentioned under Materials and Methods section of the manuscript in line 196-203.

Comment 1.2. How were they applied to the larval pans? To the surface of the water or mixed in?

Response 1.2. Upon preparing different concentrations of the fungal solutions, they were mixed in the larval pans. We have now clearly described this under Materials and Methods section in line 204-205

Comment 1.3. Observation timing why just 24 hours? Seems to be very short time

Response 1.3. Observation timing was set for 24 hours based on standard protocols of larval bioassay and other studies. (Line 245 - Reference 45-47). Also, to verify the fungal infection, SEM analysis and observation under stereo microscope was carried out which showed fungal deformities [Fig 3 and Fig 4, Fig 5]

Comment 1.4. Proper referencing

Response 1.4. Proper referencing with acknowledging original work has been done carefully.

Response to Reviewer 2:

Comment 2.1. In the abstract the objective and methodology are not clear and hard to understand so revise it carefully

Response 2.1. The abstract has been again carefully drafted with objectives, brief methodology and results for better understanding in line 20-43

Comment 2.2. Line 22: cite reference.

Response 2.2 Line 22 of previous manuscript, is in line 65 of the revised manuscript with proper referencing (Reference 8,9)

Comment 2.3. The introduction is poorly written; the current form does not meet scientific standards. Also, only 14 references have been provided, most of the sentence the citation is missing.

Write introduction with following information;

a. Mosquitoes and their health and economic burden

b. Current control methods and their drawbacks

c. biological control agent and entomopathogenic fungi mediated mosquito control and your objectives

Response 2.3. Introduction has been again drafted with mentioned headings and information along with additional information from previous manuscript with proper referencing.

a. Mosquitoes and their health and economic burden (Line 47- 71)

b. Current control methods and their drawbacks (Line 73-94)

c. biological control agent and entomopathogenic fungi mediated mosquito control and your objectives (Line 96-150)

Comment 2.4. Line 122: typo error

Response 2.4. Line 122 of previous manuscript is in line 166 of the revised manuscript with proper referencing and no typological error.

Comment 2.5. Line 131: dog biscuit? dog biscuit WHO recommended one? Why have you used this diet? Is there any special reason?

Response 2.5. Line 131 of the previous manuscript, in is line 193 of the revised manuscript with proper referencing. 

Comment 2.6. Line 144-145: how did you prepare for test concentration?

Response 2.6. Line 144-145 of previous manuscript is in line 212-218 of the revised manuscript 

Comment 2.7. Line 175: Toxicity or pathogenicity? Confirm it

Response 2.7. Line 175 of previous manuscript is in line 236 of revised manuscript. It has been corrected to larval pathogenicity test as fungi are considered as living and pathogen.

Comment 2.8. The methodological part is poorly written and contains several pieces of information.

Response 2.8. The Methodology and Materials section has been elaborated and drafted carefully again with more information in detail from line 152-270 with appropriate referencing.

Comment 2.9. The results are poorly written and data interpretation hard to follow so carefully re-edit it

Response 2.9. Results section has been drafted again with proper schematic flow and proper data interpretation for better understanding in line 273 382 with additional figures and correct referencing.

Comment 2.10. For discussion two paragraphs sufficient for a scientific paper? Did you use any new reference added in the discussion part apart from introduction and methodology? The discussion part is poorly discussed.

Response 2.10. Discussion section has been drafted again elaborately with summary of research, interpretation of results, Importance of the results procured in the research, limitation of the work and future scope with proper acknowledgement and referencing in line 384-442.

Comment 2.11. The conclusion needs more clear information; the current form is not sufficient.

Response 2.11. Conclusion has been drafted again with a clear and proper information procured from the research in line 444-456.

Comment 2.12. The reference part contains several formation and typographical errors so carefully revise it.

Response 2.12. References has been revised again with proper citation and acknowledgements using software Mendeley Cite.

Comment 2.13. Figure 1 does not provide clear and high quality of images.

Response 2.13. Figure 1: Sentinel-2 data was downloaded from USGS Earth Explorer portal (https://earthexplorer.usgs.gov/).Three bands-Band 8 (Near Infrared), Band 4 (Red) and Band 3 (Green) were used to prepare the False Colour Composite (FCC) Image. This was performed using Composite Band Tool in Arc GIS 10.2 software.

Comment 2.14. Figure 2: I cannot see any damages and any information from this figure so provide clear images.

Response 2.14. Fig 2 is now Figure 3 in line 329 with additional information and Figure 4 in line 352 and appropriate labelling.

Comment 2.15. In the SEM analysis I cannot see any entomopathogenic fungi spores and conidia but does the author see any spores?

Response 2.15. The SEM analysis (Fig 5) is in line 378 with appropriate information, additional images and labelling.

Comment 2.16 Throughout the manuscript I can see several grammatical and typographical errors so carefully fix it.

Response 2.16. The manuscript has been thoroughly revised and all grammatical and typographical errors has been fixed.

---

## [Editor Report · Decision Letter 1]

30 Jul 2024

EVALUATION OF SYNERGISTIC EFFECT OF ENTOMOPATHOGENIC FUNGI BEAUVERIA BASSIANA AND LECANICILLIUM LECANII ON THE MOSQUITO CULEX QUINQUEFASCIATUS

PONE-D-24-10887R1

Dear Dr. Kataki,

We’re pleased to inform you that your manuscript has been judged scientifically suitable for publication and will be formally accepted for publication once it meets all outstanding technical requirements.

Kind regards,

Rachid Bouharroud

Academic Editor

PLOS ONE
---

## [Editor Report · Acceptance letter]

15 Aug 2024

PONE-D-24-10887R1 

PLOS ONE

Dear Dr. Kataki, 

I'm pleased to inform you that your manuscript has been deemed suitable for publication in PLOS ONE. Congratulations! Your manuscript is now being handed over to our production team.

Kind regards, 

on behalf of

Dr. Rachid Bouharroud 

Academic Editor

PLOS ONE